# Analyzing and Leveraging Social Media Disaster Communication of Natural Hazards: Community Sentiment and Messaging Regarding the Australian 2019/20 Bushfires

Sarah Gardiner [1], Jinyan Chen [1,2], Margarida Abreu Novais [1], Karine Dupré [3,*] and J. Guy Castley [4]

[1] Griffith Institute for Tourism, Department of Tourism, Sport and Hotel Management, Griffith University, Gold Coast 4222, Australia; s.gardiner@griffith.edu.au (S.G.); jinyan.chen@polyu.edu.hk (J.C.); m.abreunovais@griffith.edu.au (M.A.N.)

[2] School of Hotel and Tourism Management, The Hong Kong Polytechnic University, Hong Kong TU428, China

[3] School of Engineering and Built Environments, Griffith University, Gold Coast 4222, Australia

[4] School of Environment and Science, Griffith University, Gold Coast 4222, Australia

* Correspondence: k.dupre@griffith.edu.au

**Abstract:** This research presents a new model based on Twitter posts and VADER algorithms to analyze social media discourse during and following a bushfire event. The case study is the Gold Coast community that experienced the first bushfire event of Australia's severe Black Summer in 2019/2020. This study aims to understand which communities and stakeholders generate and exchange information on disasters caused by natural hazards. In doing so, a new methodology to analyze social media in disaster management is presented. This model enables stakeholders to understand key message themes and community sentiment during and following the disaster, as well as the individuals and groups that shape the messaging. Three main findings emerged. Firstly, the results show that messaging volume is a proxy for the importance of the bushfires, with a clear increase during the bushfire event and a sharp decline after the event. Secondly, from a content perspective, there was a consistent negative message sentiment (even during recovery) and the need for better planning, while the links between bushfires and climate change were key message themes. Finally, it was found that politicians, broadcast media and public commentators were central influencers of social media messaging, rather than bushfire experts. This demonstrates the potential of social media to inform disaster response and recovery behavior related to natural hazards.

**Keywords:** bushfire; sentiment analysis; disaster communication; social media; Big Data

## 1. Introduction

Natural hazards, such as floods, drought, cyclones, heatwaves, and bushfires, are a global phenomenon, with some increasing in prevalence due to anthropogenically driven climate change [1]. The resulting disasters, in particular bushfires, can affect large areas of the natural environment as well as communities living in and around these affected regions [2,3]. The response to, and communication around, these events can be critical in the recovery phase following disasters [4,5]. Within this context, emergency communication concerns not only the capacity of the emergency worker who takes the initial phone call to identify the hazard level and trigger the adequate response, but also the verbal interactions, decisions and processes during the event that guide both the response and the communication about it [6]. Scholarship has shown the richness of emergency communication studies, as they not only reveal social and institutional interactions but also the level of technology, human resources, financial and other capitals, and resilience existing in a community [6].

Regarding communication about the event, it has been traditionally disseminated through reports on the disaster via broadcast media channels, such as newspapers, television and radio. However, Rafliana et al. [7] argue that a broader understanding of disaster



communication is needed beyond a one-way push-communication model using traditional channels such as broadcast media.

Increasing public access to the Internet since the 1990s and the later advent of social media platforms such as Twitter, Facebook, and YouTube in the 2000s have radically changed the ways that people access information and news [8,9], including in relation to disasters [10–12]. New communication technologies are increasingly allowing users to contribute to these information streams. However, there is also a downside. With ready access to information comes the challenge of filtering the volume of information, as well as the risk of spreading misinformation and disinformation and the challenge of distilling the truth in this information [13]. Despite these potential limitations, online social media messaging is now a central form of disaster communication [14]. Social media is increasingly used by the broadcast media, disaster management government agencies, politicians, scientific experts, and the private sector, as well as the public, to access and share emergency information pertaining to disaster events. The time currency of posts enables users to share information in real time, which can result in improved information currency and response time [15]. Accordingly, a better understanding of the role of social media messaging as a central means of communication in the modern world of disaster management communication is needed. This research intends to build upon the emerging body of knowledge that shows that social media now provides a vital platform through which communities and other stakeholders generate and exchange information on disasters caused by natural hazards. This study seeks to advance the methodology used to measure community sentiment and identify the message sender to identify which stakeholders influence messaging in the management of a disaster.

## 2. Social Media and Disaster Communication

Research suggests that social media can be employed across the three phases of disasters: before, during and after the events [15]. According to the Congressional Research Service in the United States, social media sites rank as the fourth most popular source for accessing emergency information [16]. The Australian Institute for Disaster Resilience suggests that individuals and community members, particularly community groups and local leaders, can play a critical role in disaster communications through social media to warn others about disaster events. They often have a personal connection to the location of the event and established connections with others in the impacted areas [17]. For example, users might receive disaster preparedness information and warnings via a message posted on a social media platform. It can also be used during a crisis event to send messages among various stakeholders, such as professional disaster managers and the people experiencing the disaster [18,19]. Finally, social media can be used in recovery to communicate and connect community members following a disaster [18].

Reviews of the use of social media in relation to disasters propose that social media provides an opportunity for collective behavior in response to disasters [19,20]. This enables individuals to act independently, yet at the same time connect with others through social media to create interdependent collective outcomes related to a particular social context such as a bushfire, as investigated in this study. Therefore, social media can shape how people respond to disasters and build collective knowledge. As a result, it can frame the disaster response. Zhang et al. [20] suggest that social media has three functions: (1) effective and efficient sourcing of disaster situation awareness information, (2) supporting self-organized peer-to-peer assistance and support, and (3) fostering open communication between the public and disaster management agencies. However, there is also a negative consequence of relying on social media during a disaster, as social media can also be used to spread rumors and misinformation [21]. Thus, the source of the message and the reliability of, accuracy of, and trust in social media information need to be considered [22].

This study intends to advance knowledge of the community's reactions to natural-hazard-induced disasters—with specific implications for bushfire disasters—to present a

model for analyzing community messaging in response to these events. The subsequent section discusses the collection and analysis of social media data in the context of disasters caused by natural hazards. The research questions for the study are then presented.

*The Potential Role of Twitter in Disaster Communications*

This study collects data from one of the most popular online microblogging platforms, Twitter, that is frequently used in disaster social media studies. Ogie et al. [13]'s systematic review of the disaster management literature found that 65 percent of the articles used Twitter as the data source. This platform is particularly suitable for qualitative research to measure community sentiment—that is, what the people are thinking and feeling about the disaster event—because it allows users to publish online short text messages of up to 280 characters and distribute them to other users, known as "Followers". Launched in March 2006, Twitter's popularity rapidly increased. In Australia (the location of this research), industry research found that 97 percent of respondents check social media at least once per day, including 56 percent checking it more than ten times a day, and that social media is now a primary channel for information on news and events, with 30 percent of Australians using it for this purpose [23].

The analysis of social media data, including those of Twitter, Facebook, Instagram, etc., has recently gained momentum in various research fields. These studies have analyzed social media data to explore diverse behaviors, from tourist mobility [24] to consumer retail sentiment [25] to changing health behavior [26]. Disaster research using social media data is still fairly recent, although Rasmussen and Ihlen [27] have shown that the number of studies related to social media and crises across the globe has increased dramatically over the past decade. In addition, there is an emerging body of knowledge, particularly in the information technology literature, that shows the importance of social media in disaster communication [22]. Some initial inquiry has also shown how it is possible to map people's movement and other spatial patterns during a hurricane disaster using Twitter social media data [28,29]. Ahmed [10]'s study of social media communication on disasters caused by natural hazards in Australia further advances the communication aspect of the literature, finding that social media provides a channel of communication, moral support and interaction with the rest of the world. In addition, it can provide education about the disaster and related issues. The immediacy and reach of social media mean it is also a mechanism to issue warnings and receive updates on the disaster. Ahmed [10] also found that there is agency-to-agency interaction, which assists in coordination and collaboration. Ogie et al. [13]'s systematic review of the literature exploring social media use for disaster recovery found that over half (55 percent) of the literature is based on research in the United States and the geographic spread of the literature is limited. Thus, it is evident from the expanding body of academic and industry research that social media is increasingly being used to inform, monitor and respond to disasters.

The growing role of social media in disaster communications has resulted in the decentralization of information dissemination in disaster responses, and a bottom-up rather than a top-down transfer of information on disasters is occurring, facilitated through social media communications [7]. Crowe [30] also suggests that the collective knowledge of multiple people who are experiencing the disaster and following and posting about it on social media often provides a better insight into what is happening than a single professional disaster manager or organization. Thus, there is virtual citizenship that social media provides that can assist in preparedness, response and recovery from disasters.

This study seeks to fill the gap in the literature on the role of social media messaging in disaster recovery by investigating Twitter social media posts during and following a bushfire disaster. This study also seeks to determine community views or opinions—expressed as community sentiment—to demonstrate the role that social media can play in monitoring the community response. As a result, a new methodological framework is proposed that enables stakeholders to understand the message response and themes and, accordingly, enact more effective and efficient responses than were previously possible through traditional

media channels. Advancing this understanding is important given the increasing frequency of disasters because of natural hazards, evident in countries such as Australia [31], which is the location of this study, and the growing influence of social media on disaster communications [7,13]. This paper proposes that the level of discussion of a key theme is a proxy for the importance of the issue to the community during the disaster and in its aftermath. In doing so, this paper intends to employ new methodological approaches using social media data to advance the disaster communication literature. This study specifically examines the social media response to the first bushfire event of the season that destroyed a major tourism icon, the Binna Burra Lodge, and the subsequent wide-scale Australian Black Summer Fires of 2019/20. Accordingly, the study's research questions are:

- Research question 1 (RQ1): how did the Binna Burra Lodge bushfire event, and the subsequent Black Summer Fires in Australia, influence the volume of communication on Twitter about bushfire events over time?
- Research question 2 (RQ2): what is the community sentiment revealed on Twitter during, and in the aftermath of, a disaster resulting from a bushfire event?
- Research question 3 (RQ3): what themes are central to the Twitter community's view of a disaster in the aftermath of a bushfire?
- Research question 4 (RQ4): who are the central influencers of bushfire messaging during this period?

## 3. Conceptual Framework for This Study

To understand the role of social media in disaster communications, this study proposes a hierarchical conceptual model to capture the underlying context and foundations central to our study. Figure 1 presents this model. At the first level is the acknowledgment of the timeframe for the analysis as well as the focal community and how they would respond to a disaster event. At the next level is the identification of those communication agents contributing to the disaster discourse, as well as their sentiments. Finally, at the lowest level are the individual influencers who may be central to sharing information and the themes related to such messaging.

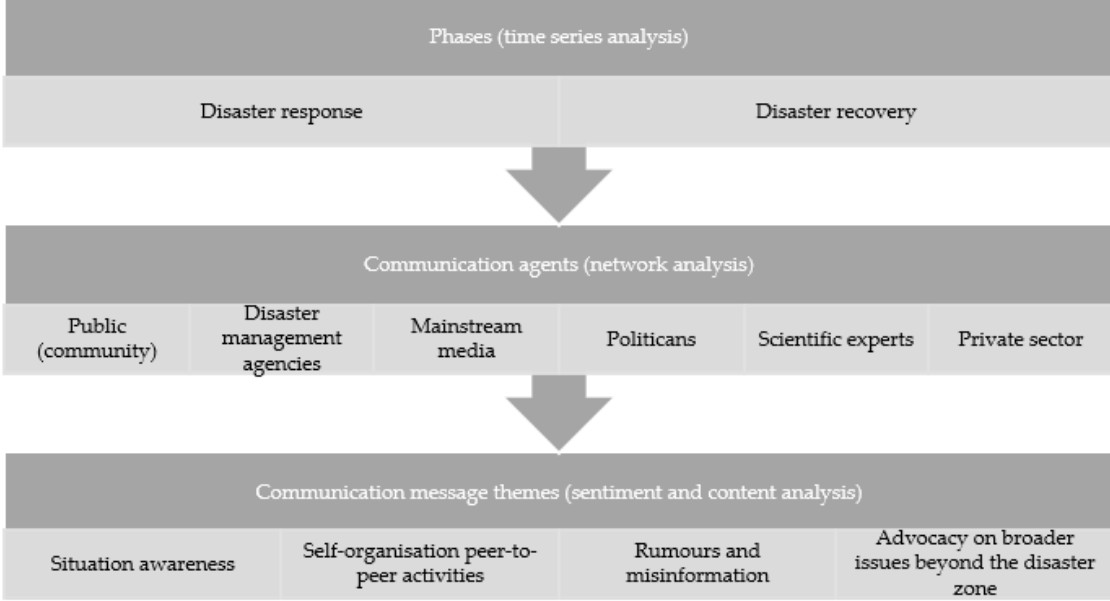

**Figure 1.** A conceptual framework highlighting the hierarchical role of social media and communication agents in disaster response and recovery.

The conceptual model also intends to outline the methodology for this study. It shows that the disaster response and recovery were analyzed using a time series analysis of

Twitter posts during and following the disaster. The agents of the communication were then investigated to determine the community sentiment of the messaging. This study extends the risk communication governance agents proposed by [7], suggesting that there are six principal communication agents, including the public (community), disaster management agencies (government), politicians (government), scientific experts, broadcast media and the private sector (i.e., organizations that are not owned or operated by a government). It should be noted that Ogie et al. [32] defined a category of "scientists and experts"; however, this study instead proposes to narrow the definition of this group to scientific experts and, therefore, categorize individuals without scientific qualifications outside of this category. This category includes self-proclaimed experts or citizen journalists [33,34] who used social media to build their reputation as an expert and network following, usually on a niche topic. These experts can gain celebrity-like status from their posting on social media. Accordingly, terms such as Twitter-famous and Instafamous (referring to social media celebrities on Instagram) have emerged [35]. Yet, Enke and Borchers [36] define social influencers as "third-party actors who have established a significant number of relevant relationships with a specific quality to and influence on organizational stakeholders through content production, content distribution, interaction, and personal appearance on the social web" [36] (p. 267). It is, therefore, possible that all categories of agents explored in this study could be potential social influencers. This definition pertains to the final stage of this analysis, which is to understand key themes in the content of the messages and the central influencers in the posting of messages. Based on the literature, four key outcomes are expected to be found in Twitter posts, including (1) gaining situation awareness, (2) self-organizing peer-to-peer activities, (3) providing a source of rumors and misinformation, and (4) serving as a form of advocacy for broader issues beyond the disaster zone, such as climate change, given the increasing public awareness and activism for this issue.

## 4. Methodology

This study used Twitter data to gain a better understanding of social media disaster communications, communication agents, and sentiments. The Twitter platform enables registered users to post micro-blogs, known as tweets, which can be read by a wider unregistered audience as well as registered Twitter users. Tweets can also be shared via the Twitter platform as well as other social media platforms, such as Facebook and Instagram, and through emails and mobile text messaging. Time series analysis of Twitter posts (related to RQ1) as well as two natural language processing techniques known as sentiment analysis (related to RQ2) and content analysis (related to RQ3) were undertaken. The study also extracted the social media user's profile information to identify the central influencers of bushfire messaging during the bushfires and in their aftermath (related to RQ4).

### 4.1. Study Context

This study involved the social media messages relating to the series of unprecedented high-severity bushfire events in Australia that began in late 2019 and continued until early 2020 named the *Black Summer Fires*. These fires impacted more than 7 million hectares of forest and woodland in eastern Australia and had a significant impact on fauna and flora as well as human life and property, as 33 people lost their lives and more than 3000 homes were destroyed [37–44].

The early-season bushfires destroyed the Binna Burra Lodge (Figure 2) during an event that was deemed catastrophic. This Lodge was a major tourism icon located in the Gold Coast hinterland, Queensland (Australia), and part of the World Heritage-listed Gondwana Rainforests of Eastern Australia. Figure 3 presents a map of the location of Binna Burra Lodge and its proximity to the city of the Gold Coast as well as Brisbane in southeast Queensland, Australia.

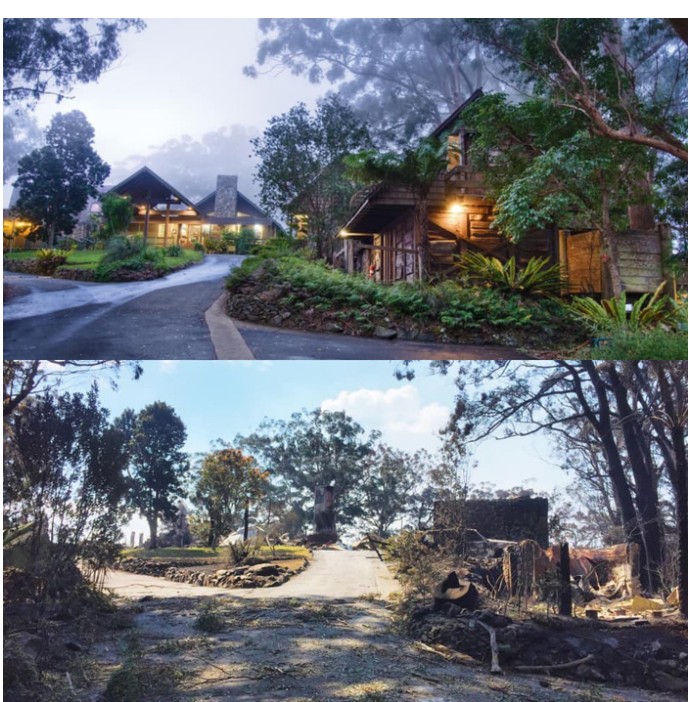

**Figure 2.** The Binna Burra Lodge before and after the bushfire. (Source: Binna Burra Lodge; photography by Leighton Pitcher.)

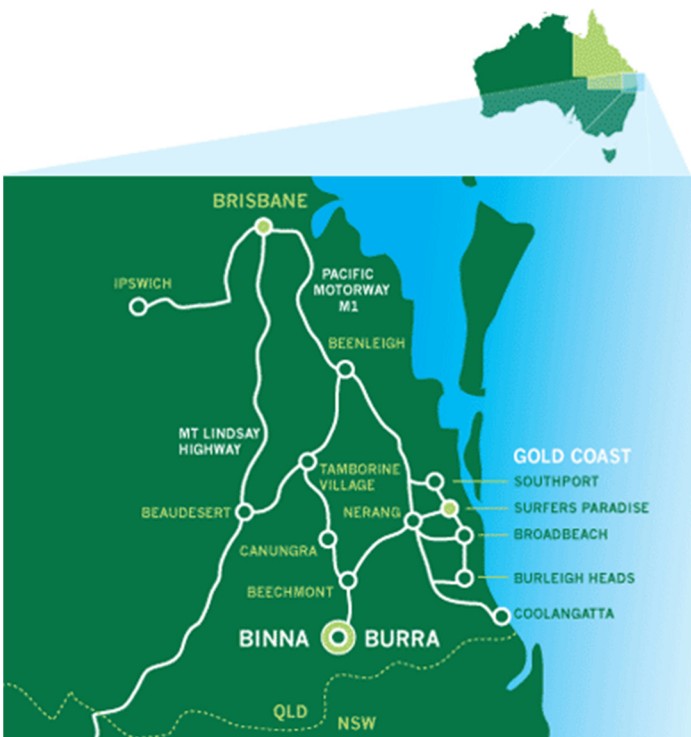

**Figure 3.** The location of Binna Burra Lodge and its proximity to the city of the Gold Coast as well as Brisbane in southeast Queensland, Australia. (Source: Binna Burra Lodge.)

*4.2. Data Collection*

The data collection process was different from using keywords and locations directly. Instead, tweets were collected first by using a bounding box around the Gold Coast (coordinates: 153.158, −28.197, 153.648, −27.776) to find all tweets sent from this region from 2019 to 2020. The next step was to filter the relevant keywords to detect the tweets

relevant to this study. Various keywords to filter the tweets were tested, such as "fire" and "wildfire", but further investigation of the tweet messages revealed that they were not related to the Binna Burra Lodge bushfire event. Only the single keyword "bushfire" was identifying relevant tweets. Since the Binna Burra Lodge bushfire occurred in September 2019, followed by extensive bushfires throughout eastern Australia that burned until March 2020, tweets were collected from August 2019 to December 2020. This way, data collection traced back to August to capture communication on Twitter before, during, and after the event. This resulted in 29,296 tweets.

Among the collected data, there are two types of tweets: One type is original tweets posted by a user. Another type is re-tweets, which stands for users who did not create original content but re-posted someone else's tweets. This is signified by RT in front of the original tweets. Based on these two types of tweets, this work included four types of analysis, with two including original and re-tweets (44,026 tweets) and two with the original tweets only (7289 tweets, after removing any tweets containing RT in front); the details are shown in Figure 4 and are stored in the document database software MongoDB.

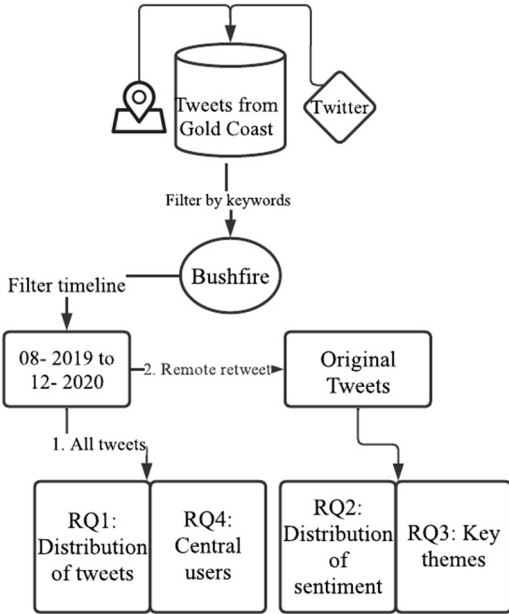

**Figure 4.** Data collection and processing framework responding to the research questions.

The full data set was applied in RQ1, which analyzes the distribution of tweets following the bushfire timeline, and RQ4, which investigates the central users. This is because the number of retweets can also show attention from the Gold Coast community and the relationship of retweeting is the fundamental factor in building user networks. On the other hand, retweets can be overwhelming, preventing the extraction of different messages from communication. Therefore, the sentiment analysis (RQ2) and extracting themes analysis (RQ3) only used the original tweets.

### 4.3. Sentiment Analysis

Sentiment analysis can be used to mine opinions from people and to understand their attitudes or emotions [45,46]. It measures sentiment expressed from the text and converts the qualitative data into quantitative scores. Although different methods have been proposed to analyze sentiment, the VADER (Valence-Aware Dictionary for Sentiment Reasoning) method has shown to be the most suitable based on the literature. VADER is a lexicon-based sentiment analysis method that has been specially designed for analyzing social media data with short text-based strings [47–49]. The pre-defined lexicon is composed of more than 7500 lexical features that were rated by professionals with a positive or negative sentiment value [46]. Those sentiment values have an intensity from −4 to 4.

For example, if the text contains the word "good", the sentiment value would be 1.9, whereas "great" is a higher value of "3.1" [46]. The full lexicon is available online at http://comp.social.gatech.edu/papers/ (accessed on: 1 January 2015).

For a better comparison, the sentiment values in this paper were normalized, with "−1 to 0" presenting negative sentiment, "0 to 1" standing for positive, and "0" being considered "neutral". The VADER only provides sentiment for English tweets, and for text written in other languages, it assigns neutral polarity.

### 4.4. Content Analysis

To see how the event triggered the community's attention, a time-series analysis of sentiment polarity was also undertaken. It helped in understanding the sentiments associated with the various bushfire events during the project timeframe. Content analysis of the Twitter posts was also conducted using Leximancer software. This machine learning text analysis software enables the semantics of the text to be revealed [50]. Leximancer detected the themes by analyzing the frequency of the co-occurring word from community discussions. Leximancer groups the concepts into themes and then ranks the themes by colors. Hot colors (red, orange) represent important themes, while cooler colors (blue, green) show the less frequent themes. Furthermore, concepts form the themes, which allows us to detect the concerns of the community.

### 4.5. Network Analysis

As explained in the section on data collection, the tweets in this study not only contain the actual text but also the relationships among users via re-tweets. Therefore, this study also adopts network analysis to identify those central influencers within the network.

To find relationships between users in the discussion, the network analysis focused on the users who: (1) tagged others; (2) were tagged in the tweet (represented by @+username in actual tweets); (3) re-tweeted others; and (4) were re-tweeted by others. These four relationships were used to create a conversation network, and based on the degree of centrality, the central users were identified.

To analyze relationships between users in this conversation network, in this study, we identified specific users using degree centrality (weighted). Degree centrality refers to the number of connections for each node in the network. A node can be defined as any element, and the connections are the lines that links these elements together [51]. For example, in Figure 5, nodes are the circles labeled with the letters A, B, and C, and the connections are the lines and arrows that connect each node. In this work, a node would be any Twitter user who was captured in this conversation, and the connections would be the four relations explained earlier.

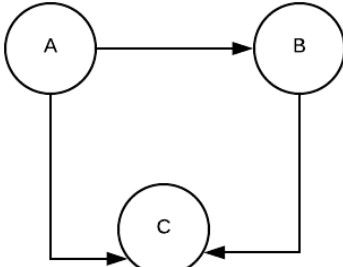

**Figure 5.** A diagrammatic representation of a directed network. The letters A, B and C are nodes (e.g. social media users), the arrows indicated information flow and connections representing measures of 'in-degree' (i.e. flow to a node) and 'out-degree' (i.e. flow from a node) centrality.

A network also can be divided into a directed network and an undirected network based on the ability to identify the sequence of connection in the former. Via Twitter, users can send messages or be tagged by others; therefore, the direction of communication can be distinguished. Therefore, we built a directed network for further analysis. In a directed

network (shown in Figure 5), there is also in-degree and out-degree centrality. In-degree centrality means that the connection with an arrow pointed at the node, which represents a user being tagged or that their tweets were re-tweeted by others. There is a weighting on the arrows based on the number of times a tweet might be re-tweeted. For example, if a tweet is posted it scores a one. If that post is retweeted 5 times, then the overall score is 6, and if it is retweeted 50 times, it carries a greater weight, that is, 50. Conversely, out-degree centrality means that the arrow comes from the node, which represents a user tagging others in their tweet or re-tweeting others' messages. An example of a directed degree network is depicted in Figure 5, where node A has zero in-degree centrality but two out-degree centralities. In the Twitter conversation network, this could mean that User A tagged/re-tweeted User B and User C, but no one tagged/retweeted User A. In the network, Nodes A, B and C each have different importance for the network. Theoretically, Node A is active in the community by frequently engaging with others, while Node C's voice has been spread well. Node B shares a similar weight in both directions, well balanced in the conversation. Applying network theory, this study identified the central users in this disaster conversation on Twitter to discuss which types of users were engaged and what kinds of messages were potentially missing from the conversation.

The degree network analysis was performed using Gephi network software. Gephi calculates a node's number of connections and groups users into communities based on the number of connections their nodes have. In this work, the communities of Twitter users were formed based on the number of connections across the four relationships stated earlier.

This work only presents the top five communities based on Twitter user interactions, indicated by five colors (red, green, purple, pink, and orange). The network analysis was further separated by year to investigate any changes amongst central users in 2019 and 2020 since the Binna Burra bushfire happened in late 2019, followed by bushfires elsewhere in Australia in 2020. The following section presents the results of the time series, sentiment, content, and network analyses of the social media data.

## 5. Results

The results of this paper are presented in four sections that relate to the study's research questions. Accordingly, findings on the volume of communication about the bushfire event over time are first presented (RQ1). Community sentiment during and in the aftermath of a bushfire (RQ2) and key themes (RQ3) that are central to the community's view of this disaster are then reported. Finally, the findings regarding the central influencers of bushfire messaging during this period are outlined (Q4).

### 5.1. The Volume of Tweets

There were a total of 7616 tweets from 2019 and a further 21,679 tweets from 2020 that mentioned "bushfire" (Figure 6). The first spike of tweets correlates with the Binna Burra bushfire event that occurred in September 2019. The number of tweets increased sharply during this period, from 27 tweets in August 2019 to 813 tweets in September 2019. The destruction of Binna Burra Lodge captured national and international media attention when an environmental not-for-profit private sector organization, Greenpeace, erected a 30 m banner on the site during the Prime Minister's visit to witness the aftermath of the bushfire, declaring this event a climate emergency [52]. Although the attention declined when the bushfires in southeast Queensland were extinguished, the discussion increased drastically in November 2019 and peaked at 10,145 in January 2020 as bushfires raged in New South Wales (NSW) and Victoria, Australia. The severity of the bushfires in December 2019 and January 2020 added to the public interest in this initial Queensland event. The image of a kangaroo against the backdrop of the fire engulfing a home, tweeted by photojournalist Matthew Abbott in January 2020, became an iconic representation of the bushfires after being featured in a story in the New York Times on 10 January 2020 [53] and was shared on Twitter. As reported in BBC News [54], as well as other news networks around the world, misleading maps showing the entire eastern coast of Australia on fire

"went viral" on social media (i.e., there was mass sharing of this content, analogous to a virus spreading). Concerns about the impacts of the bushfires on wildlife, communities, and the environment, combined with the messaging linking this event to climate change, heightened interest in bushfires. This is evident in the results of this study showing the peak of messaging in January 2020.

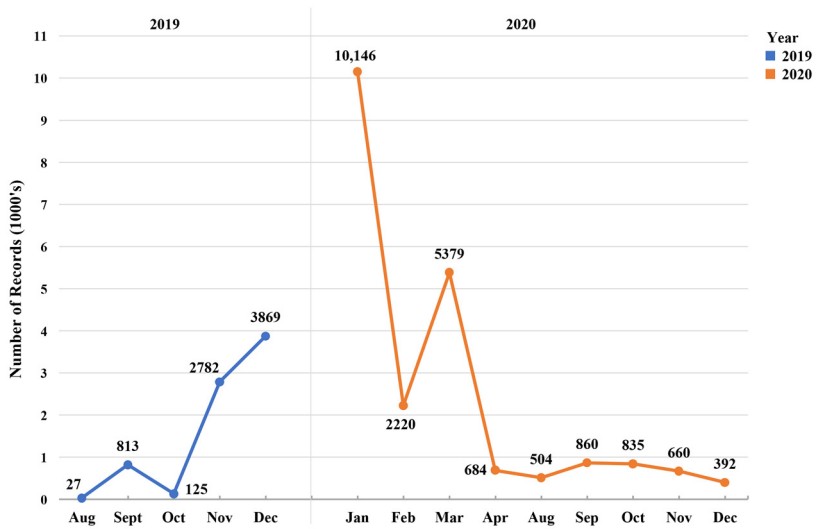

**Figure 6.** The number of tweets by month in 2019 (from August) and 2020.

### 5.2. Community Sentiment

In order to reduce the bias from massively re-tweeted information, tweets were filtered by original tweets (remove RT) to analyze sentiment to determine community sentiment before, during and following the bushfire events (related to RQ2). Figure 7 reports the number of original tweets (line charts) and the average sentiment for each month. The sentiment analysis revealed that for all original tweets related to the bushfires, sentiment values were negative. The negative sentiment was stronger in 2019 than in 2020, particularly in September, October and November 2019, due to concerns in the aftermath of the bushfires in southeast Queensland, as well as the dry climatic conditions following the preceding drought in Australia and preparedness for potential additional severe bushfires over the coming summer.

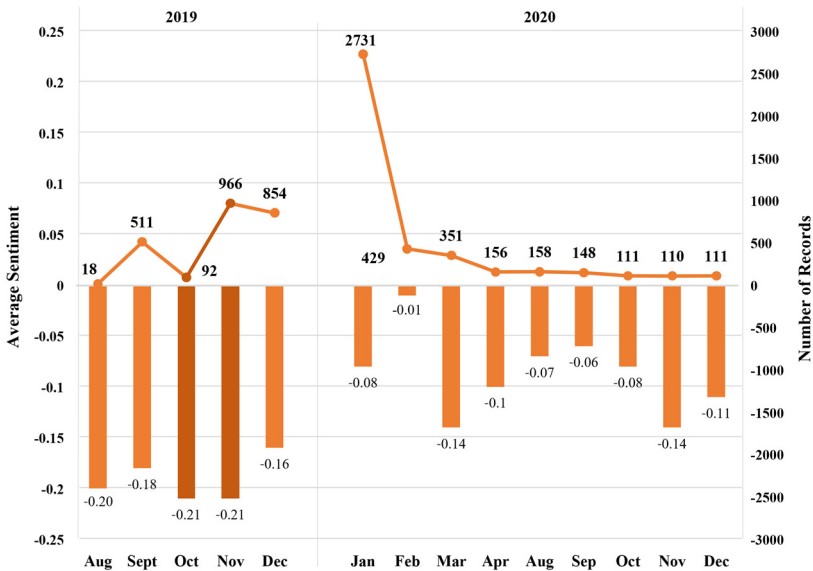

**Figure 7.** Average community sentiment of Twitter messages from the Gold Coast by month.

The first fires of the fire season gained particular traction in the media due to the destruction of the Binna Burra Lodge. The news reported that the Lodge was destroyed on 8 September 2019. Given the widespread media and government attention following the event [55], the influence of this event was further investigated by checking sentiment related to the keyword "Binna Burra" within the bushfire tweets; Binna Burra was mentioned in 87 original tweets. Figure 8 shows the distribution of the sentiment.

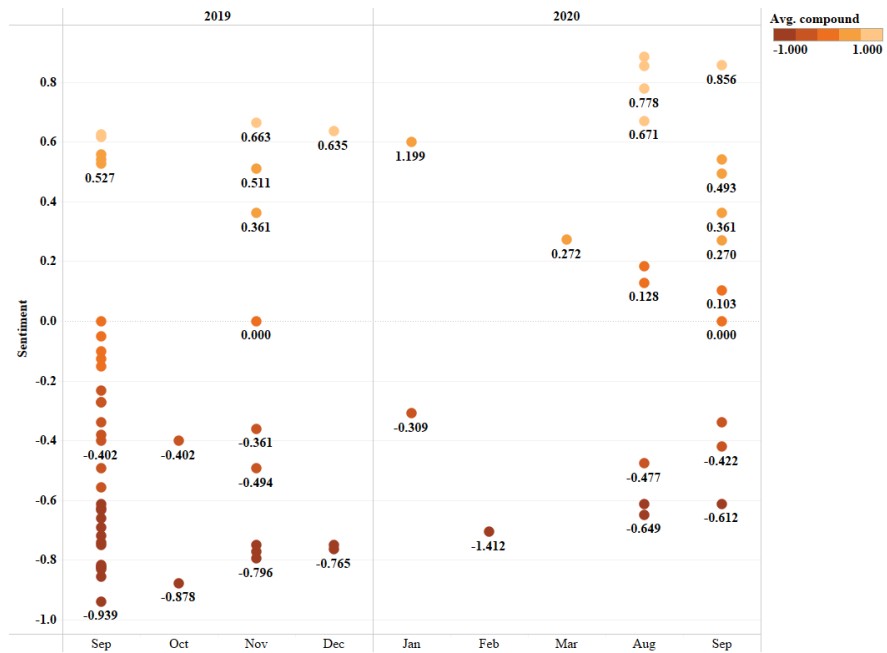

**Figure 8.** Number of posts about Binna Burra during the bushfire event.

Most Binna Burra-related tweets extracted from all bushfire tweets were from September 2019, with 45 tweets, coinciding with the destruction of Binna Burra Lodge, which also presented negative sentiment related to the destruction of the lodge. For example, one tweet mentioned that it was now too late to leave the area as driving then would be extremely dangerous. This was recognized as a negative sentiment at −0.382. Although there were negative concerns about safety issues, there was also some positive sentiment detected. This was related to staying safe, saying prayers and compliments to fire crews.

The number of tweets decreased sharply to below 10 per month in the subsequent months, showing that attention for this specific event faded quickly (refer to Figure 8). However, in November 2019, when police disclosed details about the cause of the bushfires, this attracted the attention of the community. Most of the tweets were still negative, but there was some positive sentiment, which mostly related to tweets that expressed feelings about survivors.

Another wave of attention came in August and September of 2020 when people celebrated the reopening of the lodge after one year. There were still a few negative posts during this period that related to the recalling of the devasting bushfire, but most tweets expressed positive sentiment about the recovery of the Binna Burra as a tourism destination evidenced by the return of visitors to the area.

### 5.3. Message Themes

The third phase of the analysis aimed to determine the key themes portrayed in the tweets using Leximancer for content analysis (related to RQ3). Using machine learning, Leximancer identifies concepts based on the frequency of the words in the text. The software then groups them into themes based on the co-occurrence of words in the text. The themes are color-coded, with warm colors showing important themes and less important ones in

cooler colors, as shown in Figure 9. The frequency of the words mentioned in the data is shown by the size of the grey circles next to the words, which are bushfire, ex-fire, warning, and climate change.

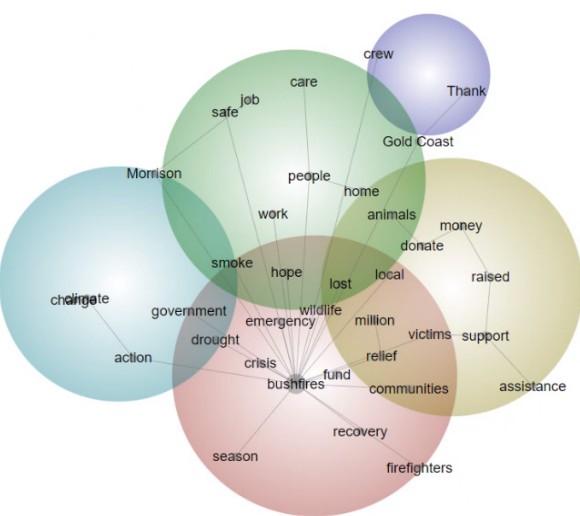

**Figure 9.** Twitter message themes relating to fire in 2019 and 2020.

Five main themes emerged from the text analysis:

The first theme (red circle in Figure 9) is centered around the bushfires keyword and some closely associated concepts such as crisis, emergency, firefighters, smoke and victims. Aspects that relate to local specificities (drought, wildlife) and recovery (including communities, relief, funds, million and hope) also emerged, with positive sentiment. The latter overlaps with the second main theme focusing on support (see the yellow circle in Figure 9), highlighting aspects such as fundraising, donating, assistance and animals.

Theme 3 (green circle) displays attention to people, their home, work, job and, like in Theme 2, animals. People expressed strong negative feelings about the threat to their homes and health. For example, one strong negative emotion expressed in the tweet was that the bushfires were affecting one's respiratory system. This overlap is explained by the preeminent agricultural characteristics of the region and the terrible impact the bushfires had on domestic livestock. This also reflects the discourse in the broadcast media and among scientific experts relating to the impacts of the fires on wildlife [40–42].

Other sentiments in Theme 3 included concerns about safety, care, hope and loss, with some of these overlapping with the first theme. Focus on specific individuals also emerged in this theme, specifically regarding the Prime Minister of Australia at that time, with the word "Morrison" (which is his surname). The use of this word also overlapped with those used in Theme 4 regarding "Climate Change" aspects. Drought and government actions with regard to climate change also emerged in the discussion. Finally, Theme 5 (shown in the purple circle in Figure 9) relates to the emergency and its impact on the fire crews and how people expressed their reactions during the bushfire.

Lastly, these tweets were further divided by positive and negative sentiment for comparison. Figure 10 shows that local issues were the main concepts for both spectrums of sentiment in the bushfire messages. Specifically, positive sentiment was found to be associated with wildlife rescues (particularly those relating to koalas) and community support and praise for firefighters. Importantly, it also appears that social media communications were used for situation awareness and self-organized peer-to-peer activities, such as fundraising efforts and support for the firefighters, as well as advocacy for broader issues beyond the disaster zone. Additionally, the connection between bushfires and climate change was one of the important topics associated with positive sentiment. On the other hand, the debate on the extent to which bushfires impact wildlife and health was the main concern for people. Finally, according to the analysis from Figures 7 and 8,

negative sentiment lasted the longest for this event and even tended to be quite severe. As shown in Figure 10, people paid much attention to the funding that was supposed to be allocated to bushfire recovery. Overall, this content analysis provides valuable information on understanding community concerns and the community's overall sentiment to improve communication in disaster management.

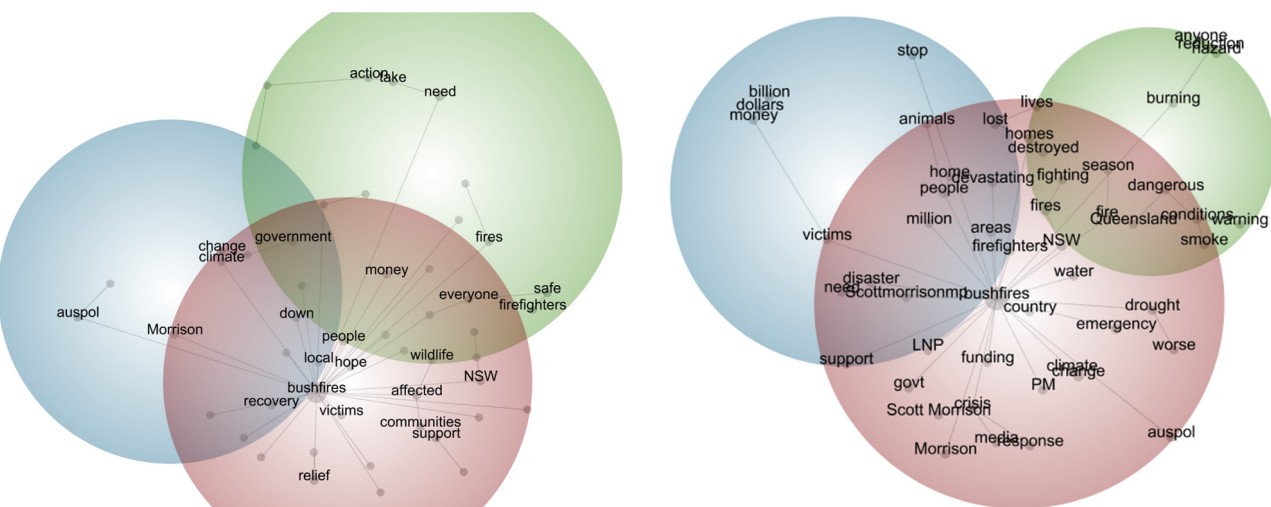

**Figure 10.** Themes of positive (**left**) and negative (**right**) sentiment towards bushfires.

*5.4. Central Influencers of Bushfire Messaging*

The fourth research question (RQ4) intended to identify the central influencers of bushfire messaging during the events. It is evident from the network analysis that there is a significant difference between the in- and out-degree user networks, which demonstrates uneven power in communication (Figure 11). Detailed information about the dominant users in the top five communities is shown in Table 1. The findings show that political leaders—including the Prime Minister of Australia, Scott Morrison, and the political Opposition Leader, Anthony Albanese (shown as AlboMP in Figure 11)—were the central figureheads tagged/retweeted in the tweets related to the bushfire in 2019. They also are the dominant users in the top two communities, where 18.26 percent of users are connected to the political Opposition Leader, Anthony Albanese. Two news media entities, SBS News and ABC News, also dominated the messaging. However, notwithstanding the influence of political leaders and the broadcast media, the dominant user in the third community was found to be a member of the public, as identified by manually checking their Twitter profile. Conversely, there was a great diversity of active users who were tagging others or retweeting posts. Not surprisingly, the local news channel (@9NEWSGoldCoast) shared the most active position, but all other users were members of the public. The user networks in 2020 showed different patterns from 2019, particularly the in-degree user networks. Detailed information about the dominant users in the top five communities is shown in Table 1.

Figure 11 and Table 1 show the members of the network whose posts received attention. Posts that were tagged or retweeted were mostly from news media entities (@canberratime and The Sydney Morning Herald—@smh), with a total of 27 percent of users belonging to the two media communities. Conversely, Figure 11 shows that the most active users who sent out messages from the first two communities were members of the public. A state government official (Queensland (Qld) Labour senator), @MurrayWatt, is the most dominant user in both the in- and out-degree networks, showing that he was very active in the topics as well as in interacting with other users (refer to Figure 11b). To identify the role of social media in disaster response and recovery, the dominant users were correlated with the framework proposed in Figure 3. Table 1 presents the users collected from the four networks to investigate their roles as communication agents.

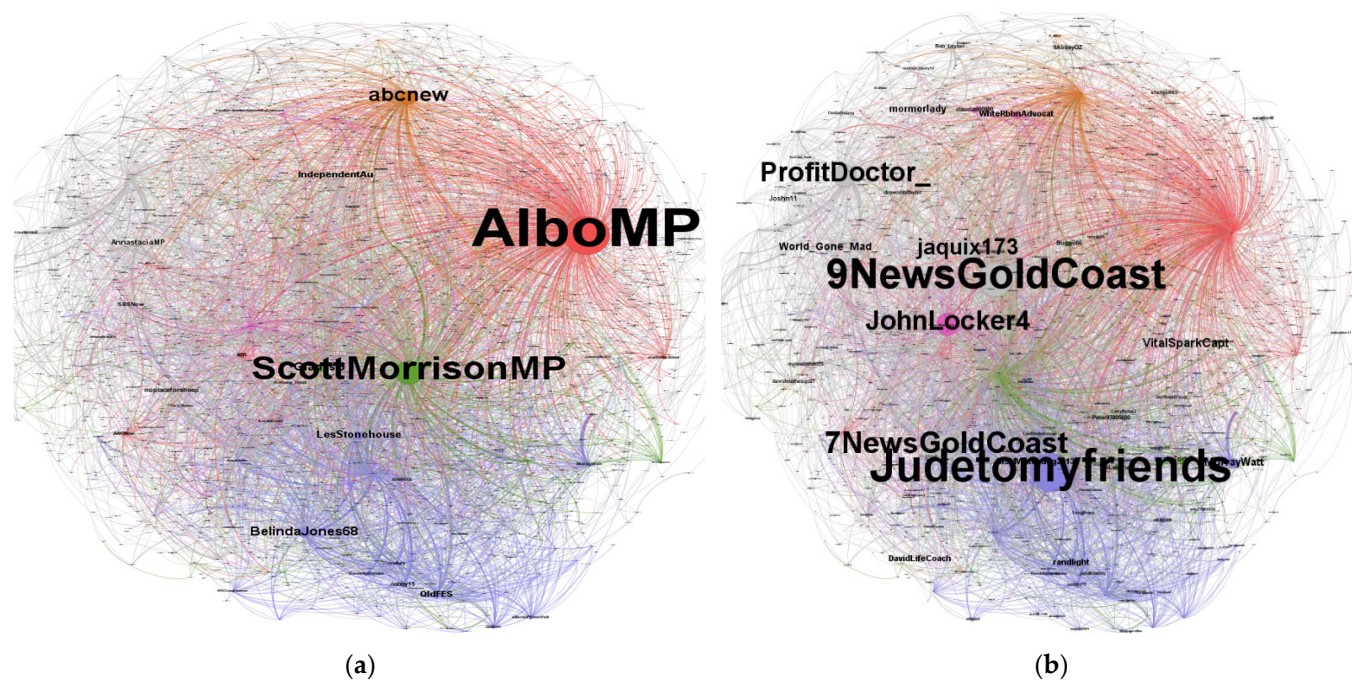

**Figure 11.** Twitter user networks in 2019 representing users as (**a**) in-degree users and (**b**) out-degree users.

**Table 1.** The role of social media in disaster response and recovery in the aftermath of bushfire events.

| Dominant User with in-Degree Centrality [1] | In-Degree Centrality | Communication Agent | Dominant User with Out-Degree Centrality [1] | Out-Degree Centrality | Account Type |
|---|---|---|---|---|---|
| @AlboMP | 576 | Politician | @9NewsGoldCoast | 178 | Broadcast media |
| @ScottMorrisonMP | 346 | Politician | @real_TomThorp | 31 | Private account |
| @BelindaJones68 | 134 | Private account | @Judetomyfriends | 200 | Private account |
| @SBSNews | 20 | Broadcast media | @JohnLocker4 | 116 | Private account |
| @abcnew | 200 | Broadcast media | @ki_sekiya | 11 | Private account |
| @canberratime | 1345 | Broadcast media | @geoffrey_payne | 37 | Private account |
| @sm (The Sydney Morning Herald) | 1429 | Broadcast media | @aconvict | 30 | Private account |
| @MurrayWatt | 169 | Politician | @MurrayWatt | 102 | Politician |
| @KristyMcBain | 113 | Politician | @jaquix173 | 545 | Private account |
| @GladysB | 59 | Politician | @Spockarama | 349 | Private account |

Note. [1]: ordering of users based on community size.

The study collected dominant users from 2019 and 2020 and clustered them based on their in- and out-degree centrality (refer to Table 1). The communication agent was also categorized (i.e., government politicians, broadcast media, and private sector accounts). Table 1 and Figure 11 show that during the bushfire events, central attention was given to politicians, who were frequently tagged or had their tweets retweeted in the community, followed by the broadcast media. In addition, one particular private account had 39,100 followers (as of February 2021), demonstrating the potential of citizen journalism and its influence on bushfire messaging.

The out-degree user network, however, was dominated by private users. Applying the network theory, the out-degree network is the users that are sending messages. During the bushfire, only one local media account (@9NewsGoldCoast) and one politician (@MurrayWatt) were active and sending messages to the community. Most of the remaining messages were sent from private accounts. Government disaster management agencies need to understand these community voices. However, it was also noted that

the government disaster management agencies were not top-listed regarding community communication for both networks in both years. Although it is commonly assumed that government disaster management agencies are supposed to be more active during events to share the right information with people or guide recovery, their voice was not evident in the communication. Instead, they relied on communicating via government politicians or broadcast news media, rather than communicating directly with the community.

## 6. Discussion

This research highlights the importance and influence of social media communication in contemporary disaster management. Most notably, this study advances the methodology of disaster communication management and evaluation by providing a new model to identify and analyze community sentiment, message themes, and influencers of messaging on social media during and following a disaster caused by a natural hazard. This new model is important because it provides a framework for researchers and policymakers to draw upon and understand the sentiment and social media conversation about a disaster in the community and pivot their response to the event and the subsequent recovery while mindful of this discourse.

The findings show that major bushfire events—beginning with the Binna Burra Lodge bushfire event in September 2019, followed by the subsequent bushfires elsewhere in Australia in the summers of 2019/20—increased the volume of communication on Twitter. Yet, this social media activity was short-term and recovery efforts did not receive the same social media attention. Despite this downturn in attention, there was a consistent negative sentiment expressed in messaging both during and in the aftermath of the bushfire disaster. One possible explanation is that the trauma that the local population on the Gold Coast felt as a result of the local bushfires (at Binna Lodge and areas surrounding the Gold Coast earlier in the bushfire period) was reflected in a feeling of empathy and psychological involvement in bushfires in other regions and that this resulted in a high volume of messaging (response) to the subsequent bushfires in other regions of Australia. The disruptive impact of a local disaster could be investigated with studies on the disorientating effects of disaster and their impact on the community psyche when other similar disasters occur, particularly during a period in close time proximity to the local disaster. Furthermore, this finding suggests that recovery expectations were not met quickly enough [56]. Another potential explanation for this finding is that the intense national and international media interest in this large-scale event heightened interest and enhanced comments and shared information on this topic. The relationship between traditional media, that is, broadcast media and new media, that is, social media is worthy of further investigation to determine ways to maximize the influence of these communication tools in disaster response and recovery and shape community behavior and sentiment in response to disaster events.

This study also undertook a content analysis of the messages to examine key themes that emerged. This investigation found that there were five primary themes in the messages. Based on prior research, it was not surprising that crisis, emergency and support for recovery are key themes and that concepts including support and people also feature in the messages. There was also a focus on the impacts of the bushfire on the ecology of the affected region, with animals and wildlife featured in the results, and the impacts of bushfires on Australian wildlife were widely reported [40–42]. Thus, social media was mostly used to gain situation awareness of the current state of the disaster, as well as for self-organization of peer-to-peer activities, such as volunteering and raising funds to support victims of the disaster and recovery efforts. These results add to the growing literature exploring the various roles of social media in a disaster, including staying connected [15], obtaining emotional support [57] and accessing vital information [13]. The need for "a plan" by the government and political leaders, most notably the Australian Prime Minister, was a central theme. This suggests that the community was using social media to heighten awareness of issues to affect government response and to boost funding and support for recovery efforts. It was also evident that social media was used to put a spotlight on

broader issues beyond the disaster zone, such as the impact of climate change on the environment. This community recognition of the connections between local disasters and global environmental challenges indicates the broader appreciation of these issues as disasters are increasingly linked to climate change on a global scale [58]. The message agenda was espoused through private accounts, showing evidence of citizen journalists and social influencers, as well as publicized through broadcast media.

Moving forward, developing tools to enable real-time monitoring and analysis of this messaging could further advance the potential of using a social media approach to disaster response and management, providing an avenue for rapid and targeted communications around such events. There is already evidence of commercial businesses using such systems. For example, Marriott International (with about 4500 hotel properties worldwide) has constructed digital geofences around some of its hotel properties to capture all publicly available social media posts from the hotel. It then analyzes and responds to these posts in real time to create a more personalized and engaging experience [59]. This technology is transferrable to disaster management and could assist in managing a disaster event in real time and responding to concerns and opportunities to support recovery in the aftermath of a disaster. Furthermore, the Internet of Things (IoT) is emerging as another tool that could further assist in managing disasters. Srirama [60] explains that "The Internet of Things (IoT) represents a comprehensive environment that consists of a large number of sensors and mediators interconnecting heterogeneous physical objects to the Internet" (p. 109). The IoT is often used in the context of establishing Smart Cities; therefore, social media could be one potential data source to manage disasters in cities and regions. Yet, there are also some challenges to this approach in terms of disaster management, as identified by Shah [61]. Some examples include fault tolerances (physical damage to infrastructure, inadequate backups, etc.), integration of large volumes, heterogenous datasets, privacy and security issues, and generating user-friendly visualizations. Therefore, while this technology remains promising, further research and development are needed to make these ambitions a reality.

In addition, the study informs practice by identifying the points of leveraging in the system. The dominance of politicians and broadcast media in the messaging is notable. Network analysis revealed that politicians, principally the Prime Minister and Opposition Leader, are central to posting messages about the bushfires and, thus, had a major influence on the messages about this disaster event. This messaging could be considered a higher level of information sharing to provide communities with updates and situation reports about the event [15]. Contrastingly, there was an absence of messaging from disaster management agencies. Considering that these agencies may be some of the first responders to the disaster and in a position to provide detailed information about local conditions, it was surprising not to see any messaging from this group, particularly since there has been a reported uptake of these platforms by emergency management organizations elsewhere [62,63]. While these disaster management agencies may have been communicating with affected communities through other channels, this finding suggests that there is a missed opportunity to utilize social media more effectively to communicate important messages and serve as a disaster management tool. It could also be argued that the messages of these agencies are communicated through the agency's political leaders. However, such messaging is often politicized, and therefore, through this lens, the accuracy and details of the message might be lost. Furthermore, the speed of message transmission could impede the effectiveness of using social media communication, particularly during a disaster event. Messaging from disaster management agencies has been criticized as being uni-directional [13]. The opportunity to share content and then receive comments from the receiver of the message in reply could provide an important tool for encouraging two-way communication and conversations about disasters and recovery efforts.

Evidence of scientific experts' influence on social messaging was not apparent from the results of this study; however, these experts may be expected to share their messages through broadcast media channels, as this may support broader contextual information-

sharing requirements. Future research could explore the infiltration of this group into the discourse in response to a disaster event related to a natural hazard. One group that did have a significant presence in messaging around the bushfires was the general public and in particular certain individuals. This marks the rise of citizen journalism, as evident in this study. Exploring ways to work with citizen journalists, who influence community sentiment and messaging on disaster response and recovery, is also an avenue to further investigation to build a more comprehensive approach to disaster communication management. Building upon the substantial literature on citizen journalism in the communications literature should be explored (e.g., [33,34]).

Replicating and building upon this methodology in other disaster locations and for other types of disasters relating to natural hazards would further advance this model. As highlighted by [13], long-term research on disaster recovery is also needed; therefore, extending the timeline to measure multiple bushfire events over an extended period (e.g., 10 years) would also advance this field of study. The analysis of photographs could also deliver a more comprehensive understanding. For instance, analyzing the visual messaging presented in an image could identify key symbols of disaster (such as the impact of a bushfire on wildlife, hence the image of the kangaroo mentioned previously), as well as alert stakeholders to the visual community discourse on the disaster. Consideration of the social media response to disasters in countries with different levels of mobile phone and social media adoption and internet capacity, most notably in developing countries, could also be beneficial to understanding how social media can be used to improve disaster response and recovery.

This study also has some limitations. For example, our research relies on a single social media platform, Twitter, to conduct data analysis. In the future, other relevant data sources could be supplied for better understanding. Furthermore, Twitter has its qualitative limitations, one example of which is being limited to 280 characters per tweet. Additionally, this study did not incorporate the images that are often associated with a tweet; this could be an additional contribution, as Sleigh et al. [64] have suggested that visuals can enhance a message and compensate for the restricted number of characters. Another limitation concerns the fact that this research did not investigate the percentage of community members who could not use social media due to lack of access to a device (e.g., phone or computer), lack of services (e.g., the telecommunication network down in some areas) or preferences/abilities (e.g., not being social-media-literate). Further research could pay attention to this specific aspect, as scholarship has shown how communication access is a crucial issue for communities [56].

Finally, the network analysis focused on the user types. If sentiment and key concepts could be detected for different communities, this would show more details of disaster communications. However, this paper's goal was to identify the social media discussions in a disaster. The analysis has covered the research questions. Future studies could work on dynamic networks to show how central users and communities change over time.

## 7. Conclusions

This study highlights the importance of and potential to use social media to communicate with the community during and following a disaster to effect positive outcomes. The social media analysis methodology presented in this paper provides a tool to advance the development of this area of disaster communications. We show the exact sentiment scores and how they change over time, providing useful information on public opinion analysis and assessing the effectiveness of recovery. Embracing new technologies and applying them to solving disaster management challenges is critical to improving the effectiveness and efficiency of disaster response and recovery. This research intends to provide a vital step in delivering on this aspiration. To provide even further insight, another study could investigate the connection between tweets to community actions.

**Author Contributions:** Conceptualization, S.G., J.C., M.A.N., K.D. and J.G.C.; methodology, S.G. and J.C.; formal analysis, J.C.; writing—original draft preparation, S.G. and J.C.; writing—review and editing, S.G., J.C., M.A.N., K.D. and J.G.C.; visualization, J.C.; supervision, S.G.; project administration, S.G.; funding acquisition, S.G. All authors have read and agreed to the published version of the manuscript.

**Funding:** Commonwealth/State Disaster Recovery Funding Arrangements 2018, Australia Grant Number: 583218-25164.

**Institutional Review Board Statement:** Not applicable.

**Informed Consent Statement:** Not applicable.

**Data Availability Statement:** Not applicable.

**Conflicts of Interest:** The authors declare no conflict of interest.

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
