# Peer review of "Analyzing and Leveraging Social Media Disaster Communication of Natural Hazards: Community Sentiment and Messaging Regarding the Australian 2019/20 Bushfires"

_societies, doi:10.3390/soc13060138_

Round 1
Reviewer 1 Report
This study analyzed disaster management through communicative approach, i.e., by using social media and tracing sentiment about Bush fires in Australia.
The study has provided a substantial broad theoretical and conceptual framework for this work.
However, there are a few aspects of the study that need further clarification.
In the sampling, keyword “bushfires” was used for the phase of the disaster. However, for control sample (i.e., before the disaster, was the same keyword used? If so, what are some conceptual expectations for this word to provide the sentiment before the disaster? Similarly, geographic location was delimited to gather tweets. It is known that typically only a small portion of tweets contain geolocation. As a result, authors either had a very stringent approach to their data collection or they bypassed it. Please clarify how data geography has been narrowed down.
Similarly, at times, studies use hashtags for their sampling. What was the presence of hashtag #bushfires in your sample and how was it treated? Similarly, what other hashtags were used and if so, were they included in sentiment analysis processing of a message? If so, how do they “skew” or shape results of sentiment analysis?
For results, authors should include examples of positive and negative sentiment tweets. Similarly, authors should include examples of messages of different stakeholders: i.e., regular users or politicians or journalists who were more likely “drive” the negative sentiment.
For theme analysis, authors should provide tweet examples for each themes.
Authors provide speculative assessment when it comes to rumors, yet no evidence is provided. Authors should run tweet account analysis to establish the percentage of non-human actors (i.e., bots in their dataset). See e.g.: https://botometer.osome.iu.edu/api Otherwise, authors should rethink how they present results on misinformation. Since no evidence is provided.
In discussion authors claim “This suggests that the community was using social media to heighten awareness of issues to affect a government response and boost funding and support for recovery efforts.” Yet their most central figures are media outlets and politicians. Those were the ones who were driving the discussion. Authors should discuss how elites perpetuate and dominate discourses online.
All in all, sentiment analysis is the weakest part of the study—i.e., it is not clear how it helps to contribute to the understanding of the phenomenon—what is the theoretical meaningfulness of this analysis?
Authors provide promising solutions for the future on how to create real-time tools for disaster management. It would be useful to engage in a broader discussion of such tools that combine IoT paradigms and social network analysis, as e.g., in :
Zelenkauskaite, A., Bessis, N., Sotiriadis, S., & Asimakopoulou, E. (2012, September). Interconnectedness of complex systems of internet of things through social network analysis for disaster management. In 2012 Fourth International Conference on Intelligent Networking and Collaborative Systems (pp. 503-508). IEEE.
Author Response
Many thanks for all your comments and suggestions.
We have tried to address all of them.

Reviewer 2 Report
Greetings, Thank you for providing the opportunity to review this manuscript. I recommend revise and resubmit. Although the work presented is not novel or new and does not, currently, showcase a contribution to the field, there is potential. Abstract This research presents a new model to analyze social media discourse through examining Twitter posts during and following Australia’s severe Black Summer bushfires in 2019/2020 from a community on the Gold Coast that experienced the first bushfire event that season. The study proposes that messaging volume is a proxy for the importance of the bushfires and found increases during the bushfire event but declines sharply following the event. However, there was a consistent negative message sentiment. The need for better planning and the links between bushfires and climate change were key message themes. Politicians, broadcast media, and public commentators were central influencers on social media messaging, and this demonstrates the potential of social media to inform disaster response and recovery behavior related to natural hazards. Questions/Aspects I want answered after reading the abstract:- What is this new model?
- Is the methodology for Twitter analysis detailed out?
- Is the context provided for the bushfires and the community being studied?
- An increase in twitter messages during a crisis is not a novel finding. What is novel about this study? How does it contribute to the fields of crisis communication/risk communication?
- How were messages deemed negative?
- What is the overall goal of this project? The abstract gives a superficial overview.
Author Response
Thank you for your comments and suggestions. We have tried to address all of them.

Round 2
Reviewer 2 Report
Great work updating the manuscript.